# Mining Amphibian and Insect Transcriptomes for Antimicrobial Peptide Sequences with rAMPage

**DOI:** 10.3390/antibiotics11070952

**Published:** 2022-07-15

**Authors:** Diana Lin, Darcy Sutherland, Sambina Islam Aninta, Nathan Louie, Ka Ming Nip, Chenkai Li, Anat Yanai, Lauren Coombe, René L. Warren, Caren C. Helbing, Linda M. N. Hoang, Inanc Birol

**Affiliations:** 1Canada’s Michael Smith Genome Sciences Centre at BC Cancer, Vancouver, BC V5Z 4S6, Canada; dlin@bcgsc.ca (D.L.); dsutherland@bcgsc.ca (D.S.); saninta@bcgsc.ca (S.I.A.); nlouie@bcgsc.ca (N.L.); kmnip@bcgsc.ca (K.M.N.); cli@bcgsc.ca (C.L.); ayanai@bcgsc.ca (A.Y.); lcoombe@bcgsc.ca (L.C.); rwarren@bcgsc.ca (R.L.W.); 2British Columbia Centre for Disease Control, Public Health Laboratory, Vancouver, BC V6Z R4R, Canada; linda.hoang@bccdc.ca; 3Department of Pathology and Laboratory Medicine, University of British Columbia, Vancouver, BC V6T 1Z4, Canada; 4Bioinformatics Graduate Program, University of British Columbia, Vancouver, BC V6T 1Z4, Canada; 5Department of Biochemistry and Microbiology, University of Victoria, Victoria, BC V8P 5C2, Canada; chelbing@uvic.ca

**Keywords:** antimicrobial peptide, AMP discovery, genome mining, antimicrobial resistance

## Abstract

Antibiotic resistance is a global health crisis increasing in prevalence every day. To combat this crisis, alternative antimicrobial therapeutics are urgently needed. Antimicrobial peptides (AMPs), a family of short defense proteins, are produced naturally by all organisms and hold great potential as effective alternatives to small molecule antibiotics. Here, we present rAMPage, a scalable bioinformatics discovery platform for identifying AMP sequences from RNA sequencing (RNA-seq) datasets. In our study, we demonstrate the utility and scalability of rAMPage, running it on 84 publicly available RNA-seq datasets from 75 amphibian and insect species—species known to have rich AMP repertoires. Across these datasets, we identified 1137 putative AMPs, 1024 of which were deemed novel by a homology search in cataloged AMPs in public databases. We selected 21 peptide sequences from this set for antimicrobial susceptibility testing against *Escherichia coli* and *Staphylococcus aureus* and observed that seven of them have high antimicrobial activity. Our study illustrates how in silico methods such as rAMPage can enable the fast and efficient discovery of novel antimicrobial peptides as an effective first step in the strenuous process of antimicrobial drug development.

## 1. Introduction

Due in large part to the overuse and misuse of antibiotics, the prevalence of multidrug-resistant bacteria is rapidly growing at a rate that cannot be matched by antibiotic discovery efforts [1]. As a consequence, the world is currently in an arms race and is at the cusp of a post-antibiotic era [1]. The slow pace of new antibiotic drug discovery, development, and regulation, combined with the accelerated emergence of resistance to existing antibiotics creates what is referred to as the “discovery void” [2]. This gap between discovery and emergence of resistance highlights an urgency to develop new antimicrobial therapeutics. One such alternative is formulations based on the antimicrobial peptides (AMPs) [3].

AMPs are short amphipathic host defense peptides that are produced in all multicellular organisms as part of the innate immune system [3]. Many AMPs operate through nonspecific mechanisms [4], such as direct electrostatic interactions with the cell membrane and immunomodulation [3], allowing for a broad spectrum of efficacy against bacteria [5], viruses [6], and fungi [7]. Furthermore, pathogens develop a slower rate of resistance to AMPs compared to conventional antibiotics [8]. It is these qualities that position AMPs as attractive alternatives to conventional antibiotics [9].

AMPs are often produced as precursor peptides within cells that consist of an N-terminal signal peptide, followed by an acidic pro-sequence, and a C-terminal basic bioactive mature peptide sequence [3]. The acidic pro-sequence neutralizes the basic mature peptide to keep the AMP in its inactive form and the signal peptide and acidic pro-sequence together are referred to as the prepro domain [3]. AMPs are then activated by proteolytic cleavage of the prepro sequence and the release of the mature peptide [3]. While the signal peptide is often highly conserved, the acidic pro-sequence and mature AMP can be quite variable [10]. However, there is evidence that the prepro sequence can vary across different organisms [3] and even within organisms [11].

Past research has shown that amphibians, such as the American bullfrog *Rana [Lithobates] catesbeiana*, possess a rich diversity of AMPs due to their aquatic and terrestrial life cycle, where the species encounter a wide spectrum of pathogens in these two environments [11]. In amphibians, AMPs such as ranatuerin are secreted at the skin surface upon pathogen exposure and can also stimulate an adaptive immune response [12]. In contrast, insects lack a sophisticated adaptive immune system and yet are highly tolerant to bacterial infection [13,14]. This may be due to the production of AMPs by the innate immune system [14]. In insects, AMPs are found in venom or salivary gland secretions. For example, melittin, a 26 amino acid (AA) peptide is the main component of honeybee venom [13]. While there are many known amphibian AMPs, there are far fewer known insect AMPs. Amphibian AMPs have their own designated database of 1923 peptides in the Database of Anuran Defense Peptides (DADP) [15]. Additionally, they also comprise 34% (1128 sequences) of the curated Antimicrobial Peptide Database 3 (APD3) [16]. Insect AMPs, however, only contribute 10% (325 sequences) in APD3, despite being the next largest non-mammalian classification. Better characterization of these AMP arsenals holds great potential in aiding the discovery of novel AMPs.

Because most AMPs under therapeutic investigation are derived from naturally occurring AMPs in various organisms [2], effective methods to discover natural AMPs would expand the number of potential candidates. Current wet lab screening protocols consist of extraction, isolation, and purification of AMPs through laborious methods such as the collection of skin secretions followed by liquid chromatography and sequence identification using mass spectrometry [17,18,19,20,21]. However, these protocols are costly, time-consuming, and expertise intensive. To resolve this, a scalable, rapid, high throughput in silico methodology built on genomics technologies and able to mine RNA sequencing (RNA-seq) datasets, would greatly aid in the discovery of AMPs funneling into drug development and enhancement processes. There are in silico AMP discovery methodologies presented in earlier studies [22,23,24,25,26], most of which start with processed data such as assembled genomic or protein sequences. Additionally, there are several state-of-the-art tools that perform AMP prediction [27]. Because AMP precursor genes have conserved sequence characteristics, these properties can be leveraged for filtering, and their inferred mature products can be classified as an AMP or not using machine learning methods. With the current unprecedented expansion of data generation and large amounts of sequencing data available in public repositories [28], there exists a rich untapped resource for AMP discovery.

To help fill the antibiotic discovery void, we offer rAMPage: Rapid Antimicrobial Peptide Annotation and Gene Estimation, a homology-based AMP discovery pipeline to mine for putative AMP sequences in publicly available genomic resources. To classify AMP sequences, rAMPage employs AMPlify [27], an attentive deep learning model. Currently, existing AMP databases, (e.g., APD3, DADP) contain less than 4000 validated nonredundant AMP sequences in total. In comparison, we have found over 1000 putative mature AMPs in the present study, with the potential to discover thousands more. Realizing the full potential of such pipelines would require the synthesis and validation of AMP candidates. Herein, we report our results on a select list of 21 peptides we detected using rAMPage.

## 2. Results

### 2.1. Identification of Putative AMPs

Using rAMPage, we assembled ~53 million transcripts from 84 RNA-seq datasets derived from the transcriptomes of 38 amphibian (33 frogs, five toads; anurans) and 37 hymenopteran insect (eight ants, five bees, 24 wasps) species and flagged 203,758 candidate peptide sequences to be classified (Figure 1). To select a list of high-confidence putative AMPs, we collapsed duplicates from multiple samples and applied three filters: AMPlify prediction score, peptide charge, and peptide length to obtain 1137 peptide sequences. Of these, 795 originate from amphibians, and 342 from insects. Running rAMPage on all 84 datasets took one week, with all datasets (comprising < 1 billion reads) taking less than 24 h (see Appendix A for details on the computational platform and resource usage statistics).

For each sequence, AMPlify [27] reports a prediction score *s* from 0 to 80, where *s* is a log-transformation of the AMPlify probability score *p*
*s* = −10 log_10_(1 − *p*),(1)
and 80 represents the highest confidence.

We note that the training data set for the AMPlify model had an over-representation of AMPs from amphibian species [27]; hence, it is biased towards assigning higher scores for amphibian AMPs. To compensate, we have applied separate score cut-offs for the two groups: 10 for amphibians and 7 for insects. Since the majority of AMPs are positively charged, a net charge threshold of ≥+2 was applied. As for length, we filtered for sequences that are ≤30 AA, because shorter peptides are more cost-effective to synthesize for downstream validation studies. Appendix A shows that the length filter used is the most restrictive filter of the three, with only 4.28% and 1.45% of the sequences for amphibians and insects, respectively, meeting this criterion.

Score, charge, length distributions, and AA compositions of the 1137 putative AMPs are characterized in Appendix A. From this set, 21 AMPs were selected for synthesis and validation, using three prioritization strategies: “Species Count”, “Insect Peptide”, and “AMPlify Score” (see Section 4). The peptides have been named after the species they were discovered from (Appendix A), then numbered in order using their AMPlify scores.

### 2.2. Antimicrobial Susceptibility Testing (AST) Results

A total of 21 of the 1137 putative AMPs (Appendix A) were synthesized (Genscript Biotech, Piscataway, NJ, USA) and tested for their antimicrobial activity against *Escherichia coli* ATCC 25922 and *Staphylococcus aureus* ATCC 29213 in a minimum of three independent experiments (see Appendix A for a full set of experimental results). In these antimicrobial susceptibility tests, AMP activity was assessed using two metrics: minimum inhibitory and bactericidal concentrations (MIC and MBC, respectively). Lower MIC and MBC values are desirable as they indicate that lower AMP concentrations are sufficient for inhibitory or bactericidal activity, respectively. AMP toxicity was measured by HC_50_ hemolytic concentration values—the concentration required to lyse ≥ 50% of porcine red blood cells. In contrast to MIC/MBC assays, it is desirable to have higher HC_50_ values. All 21 putative AMPs exhibited minimal to no hemolytic activity with HC_50_ values of 64 μg/mL or higher.

Of these 21 putative AMPs, three displayed moderate activity (MIC and MBC in the range 8–16 μg/mL) and four displayed high activity (≤4 μg/mL) against *E. coli* and/or *S. aureus*, all with minimal hemolytic activity, as shown in Figure 2. The characteristics of these seven sequences are described in Table 1. All seven AMPs with moderate to high antimicrobial activity have AMPlify scores greater than 25.

### 2.3. Novelty of Discovered AMPs

To assess if the putative AMPs discovered using rAMPage are novel, a BLASTp [29] (basic local alignment search tool) protein search was performed using the 1137 sequences that met our selection criteria. Of these, 1024 sequences are reported as novel, providing no antimicrobial characterization or exact match (sequence identity = 100%; query coverage = 100%) within the NCBI non-redundant protein database [29]. The novelty analysis results for the seven moderately to highly active AMPs are presented in Table 2. Four of the queried putative AMPs (AmMa1, OdMa12, PeNi10, and PeNi14) are novel in sequence, aligning with high sequence identity (≥90%) to existing NCBI annotations [29]. Two putative AMPs (PeNi11 and TeBi1) are known and published AMPs, aligning with 100% sequence identity of the precursor protein and across the prepro and mature regions. One putative AMP (TeRu4) aligns with high sequence identity to an uncharacterized protein in the NCBI non-redundant protein database.

AmMa1, derived from the Mouping sucker frog, *A. mantzorum*, aligned with 97% sequence identity to Palustrin-2GN3 [30] from a species of the same genus, *A. granulosus*, differing only by two AA in the mature region (Appendix A). Similarly, OdMa12, found in the green odorous frog, *O. margaretae*, aligned with 98% sequence identity to odorranain-F2 [31] from a species of the same genus, *O. grahami*, differing only by one AA in the mature region (Appendix A). While these two sequences (AmMa1 and OdMa2) are very similar to known sequences, we have additionally discovered each of them in a different species of the same genus.

PeNi10 was detected in the dark-spotted frog *P. nigromaculatus*, and aligned with 82% identity to pelophylaxin-1 [32] from a species of the same genus, *P. fukienensis* (Appendix A). We also identified PeNi10 in four other species of frogs: *L. boringii*, *P. megacephalus*, *R. dennysi*, *R. omeimontis* (Appendix A). Although the PeNi10 precursor aligns best to pelophylaxin-1, the mature region aligns with complete sequence identity to ranatuerin-2N (unpublished).

PeNi14, also derived from the dark-spotted frog, *P. nigromaculatus*, aligned with 90% sequence identity to palustrin-2HB1 [33] from a species of the same genus, *P. hubeiensis* (Appendix A). PeNi14 was also detected in three other species of frogs: *B. gargarizans*, *P. megacephalus*, *R. omeimontis* (Appendix A).

Originating from the dark-spotted frog, *P. nigromaculatus*, PeNi11 aligned with 100% sequence identity to pelophylaxin-1 [32] from a species of the same genus, *P. fukienensis*, meaning it is identical to a known AMP precursor (Appendix A). However, in addition to *P. nigromaculatus*, we also detected PeNi11 in four other species of frogs: *L. boringii*, *P. megacephalus*, *R. dennysi*, *R. omeimontis* (Appendix A).

Found in the venom of tramp ant, *T. bicarinatum*, TeBi1 aligned with 100% sequence identity with bicarinalin [34,35] of the same species (Appendix A). In the case of TeBi1, its precursor was partial on the 5′ end, accounting for no alignments in the prepro sequence.

TeRu4, discovered in the brain of the small myrmicine ant, *T. rugatulus,* aligned with 100% sequence identity to an uncharacterized protein [36] from a species of the same genus, *T. longispinosus* (Appendix A). While TeRu4 is not a novel protein, it is a novel mature AMP as it has not been previously characterized to have antimicrobial properties.

Additional annotation of the seven bioactive peptides (five amphibians, two insects) can be found in Appendix A. The underrepresentation of insect AMPs in the literature, compared to amphibians, is further demonstrated here; while the amphibian peptides have been annotated with “frog antimicrobial peptide” domains in both InterProScan [37] and Pfam [38], the insect sequences have no protein family annotations. Appendix A illustrates the sequence identity between AMPs identified by rAMPage and known AMPs for amphibian and insect AMPs. Although the majority of putative AMPs from rAMPage were novel sequences, previously reported AMP sequences were also identified and are a good demonstration and internal validation of the robustness of this methodology.

## 3. Discussion

Using rAMPage, we analyzed 84 RNA-seq datasets of 38 amphibian and 37 insect species to discover 1137 putative AMPs, 1024 of which are novel. In the present study we report our validation results on 21 putative AMPs, with over 1000 additional peptide sequences left to investigate. This list is by no means exhaustive; adjusting the described filtering parameters may yield thousands more discoveries (Appendix A). Further, the rAMPage pipeline can be readily used on other transcriptome sequencing datasets, though this might call for modifications in experimental designs. For instance, in the case of bacterial RNA-seq datasets with reduced post-transcriptional polyadenylation, RNA-seq data from rRNA depleted libraries would be recommended as input for the pipeline, as opposed to data from poly(A) enriched libraries [39,40].

While the sensitivity (proportion of reference AMPs captured by the three putative AMP filters) of rAMPage is <50% (Appendix A) with the default filtering thresholds, the filters are implemented to select for high confidence predictions that are also easier and more cost-effective to synthesize for validation. However, as more putative AMPs are discovered and the number of reference AMPs increase in public databases, the rAMPage filters can be adjusted accordingly to report more novel AMPs.

Although rAMPage captures most putative AMPs in their complete mature form, their associated precursor sequences may be incomplete, as shown using multiple sequence alignments with Clustal Omega v1.2.4 [41] (Appendix A). However, most partial transcripts are missing sequence on the 5′ end. Therefore, while the AMP precursors may be partial, the mature AMPs at the C-termini are more likely to be complete, thereby still detectable by rAMPage.

Because progress is rapid in bioinformatics, rAMPage is designed to be flexible as new technologies are developed. The pipeline is implemented as a Makefile with each step as a separate target, making the pipeline modular and providing analysis checkpoints. The tools for each step can be substituted with newer/improved tools if needed. Similarly, the pipeline is versatile and can be adapted for other sequencing technologies, for instance by assembling RNA/cDNA long reads from Pacific Biosciences of California (Menlo Park, CA, USA) or Oxford Nanopore Technologies Ltd. (Oxford, UK) instruments.

Recently, our group released AMPlify and compared its performance to other state-of-the-art tools for AMP prediction [27]. Other machine learning methods included iAMPpred [42], iAMP-2L [43], AMP Scanner Vr. 2 [44], with AMPlify outperforming all previously described AMP prediction tools in metrics of accuracy, sensitivity, and specificity [27]. For this reason, rAMPage employs AMPlify as its AMP prediction step, and will continue to until it is surpassed in performance. Machine learning in AMP discovery is a dynamic study, ranging from AMP sequence prediction and structure classification to de novo AMP sequence generation and design [45,46,47]. While there are existing methods to mine protein databases [48,49], rAMPage is an all-in-one tool to mine next-generation sequencing data directly from reads to AMP prediction.

While rAMPage can find a substantial number of putative AMPs, its main limitation lies in the fact that it uses homology-based sequence selection and machine learning-based sequence classification steps. These two steps are limited by the quantity and quality of data currently available for training the tools. The homology-based step of rAMPage would be less sensitive when there are more divergent signal sequences in the precursor genes. Similarly, the sequence classification engine in the pipeline, AMPlify, may be biased by known (and limited) classes of AMPs in the databases. However, this limitation is not restricted to only AMPlify, but all approaches dependent on AMP databases for training data sets [48,49,50].

Despite these limitations, which are expected to resolve over time as curated AMP sequence databases grow, a sizeable number (>1000 from 84 RNA-seq datasets) of AMPs were reported by the pipeline with the filters described herein. In the tested set of 21 peptides, seven demonstrated antimicrobial activity against a defined set of bacteria in vitro and 15 did not. We note that AST experiments can assess activity against the tested pathogens but cannot rule in or out an activity against other targets. Further, AMPs have multiple modes of action, and the AST protocol used in our study only validates direct action and does not test the putative immunomodulatory effects of these peptides, for instance. Of the seven active putative AMPs, three were moderately active, and all three are expressed in multiple amphibian species, potentially signaling the evolutionary significance of these AMPs.

An AMP of particular interest in the present study is TeRu4, due to its novelty and specificity in bioactivity. The precursor sequence of TeRu4 is 234 AA long, indicating that TeRu4 may be a multi-functional protein, such as a histone whose subsequence includes antimicrobial properties [51]. Additionally, TeRu4 showed a 36.84% sequence similarity to the spaetzle protein from the fruit fly *Drosophila melanogaster*, a protein in the insect Toll pathway, which triggers AMP production [52]. TeRu4 is also the most specific of the active putative AMPs we tested. While all the other active peptides tested are active against both *E. coli* and *S. aureus*, TeRu4 is active only against *E. coli*, a Gram-negative bacterium. This specificity may indicate a unique mechanism of action.

Despite the great promise of discovering putative AMPs with rAMPage, AMP-based drug development still faces some biological challenges, such as peptide stability and bacterial resistance. AMPs in their mature form are considered more unstable and more easily degraded by proteases. While synthesizing precursors for testing would increase stability, doing so would drive up the cost of synthesis using conventional synthetic chemistry methods. Although resistance to AMPs emerges at a slower rate compared to resistance to antibiotics, bacteria may develop resistance to AMPs through surface remodeling, modulation of AMP gene expression, proteolytic degradation, trapping, efflux pumps, and biofilms [4,53,54,55]. To combat specific mechanisms of resistance, targeted AMP discovery methods are being developed. A method to discover AMPs with anti-biofilm activity is described in a preprint [26], and a curated 3D structural and functional repository of AMPs relevant to biofilm studies called B-AMP was recently published [26]. Finding solutions to these and other challenges in developing AMPs as replacements for conventional small molecule antibiotics is an active field of research [56,57,58].

## 4. Materials and Methods

rAMPage is an AMP discovery pipeline that takes short RNA-seq reads as input, and outputs candidate putative AMPs for wet lab validation. Since it is a homology-based method to select a list of candidates for classification, a set of reference AMPs is required. Here, we describe how input datasets and reference AMPs are collated, as well as each step of rAMPage.

### 4.1. Collating Input RNA-Seq Datasets

The RNA-seq reads from 38 amphibian and 37 insect species were downloaded from the Sequence Read Archive (SRA) [59] using fasterq-dump v2.10.5 (http://ncbi.github.io/sra-tools/, accessed on 4 November 2019) from the NCBI SRA Tool Kit. Analyzing RNA-seq (transcriptomic) reads enables the discovery of expressed putative AMPs. Because some RNA-seq experiments were conducted with multiple tissues or treatments, there are 75 species in total, but 84 datasets are shown in Appendix A.

### 4.2. Collating Reference AMP Datasets

A set of 3306 AMP sequences were collated from two high-quality AMP databases: the Database of Anuran Defense Peptides (DADP; http://split4.pmfst.hr/dadp/, accessed on 6 December 2018) [15] and the Antimicrobial Peptide Database 3 (APD3; https://aps.unmc.edu, accessed on 14 September 2020) [16]. These databases are highly curated, where sequences have been validated for efficacy. To complement this list, 3835 precursor and mature AMP sequences of amphibian and insect origin were downloaded from the NCBI non-redundant (nr) protein database [29]. These sequences are less curated, including partial sequences and sequences with only in silico prediction, etc., accounting for the difference between numbers from DADP/APD3 and NCBI in Appendix A.

### 4.3. rAMPage Pipeline

rAMPage is implemented as a Makefile and written in bash, Python3, and R. It is publicly available on GitHub (https://github.com/bcgsc/rAMPage, v1.0 accessed on 14 February 2021). The pipeline was tested for the dependencies listed in Appendix A, and is highly customizable, with its major parameter options listed in Appendix A. Command and parameters for each step can be found in Appendix A. A flowchart of the rAMPage pipeline is shown in Figure 3.

Because the datasets used for rAMPage originate from publicly available genomic resources and we have no control over the experimental design or protocols used, we performed rigorous quality control. The RNA-seq reads were trimmed to remove adapter sequences using fastp v0.20.0 [60], which does not require the adapter sequences to be known, and instead infers adapter sequences from sequence overlaps between reads. This is particularly convenient when dealing with multiple datasets that possibly have different sequencing protocols.

To assemble the RNA-seq reads into transcripts we used RNA-Bloom v1.3.1 [61], a de novo transcriptome assembler that works with single and paired-end reads. RNA-Bloom is able to assemble transcriptomes without a reference but also allows for reference-guided assembly if a reference is available. It also allows for multi-sample pooling, where, for instance, reads describing multiple tissues from the same individual or different treatments for the same species are assembled together while retaining the tissues or treatment specificity of assembled transcripts.

We note that the transcripts with a smaller number of reads have less reconstruction evidence; thus, assembled sequences with lower measured expression levels may be enriched for misassemblies. To exclude such sequences from downstream analysis, we used Salmon v1.3.0 [62] to quantify assembled transcript expression levels, and filtered out transcripts with less than 1 TPM (transcripts per million) expression.

To obtain translated peptide sequences from the transcripts, TransDecoder v5.5.0 [63] was used to conduct an in silico six-frame open reading frame (ORF) translation, and ORFs that are at least 50 AA were selected for downstream analysis. In the case of nesting ORFs, the longest ORF was chosen.

To select putative AMP precursors from this vast pool of assembled and translated sequences, we conducted a homology search against our curated reference AMP dataset (Appendix A) using HMMER v3.3.1 [64] and assigned an Expect (E) value to every sequence. The E-value describes the number of hits expected by chance when searching a database of a particular size [65]. Sequences that share a certain degree of identity, with E-values of less than 10^−5^, were selected as putative AMP precursors.

These putative precursor (or partial precursor) sequences were then cleaved in silico using ProP v1.0c [66] to obtain putative mature AMP sequences, to be further classified. However, cleavage prediction tools only predict where the cleavage occurs, not what each resulting cleaved peptide represents, and the AMP precursor organization shows inter- and intra-species variability [13,67,68]. While amphibian AMPs are typically cleaved at a lysine–arginine (KR) motif and their precursor structure follows a conserved structure (prepro sequence containing acidic AA residues and a mature bioactive AMP) [67], insect AMPs are typically cleaved at an RXXR motif (two arginine residues surrounding two optional AA) and the precursor structure is not always conserved [68]. Insect AMPs are more variable in structure [13], increasing the difficulty in identifying the putative mature peptide. This difficulty is especially present in precursor structures with multiple acidic regions (UniProtKB P54684.1) or multiple bioactive regions (UniProtKB P35581.1). In such multi-peptide precursors, it is unclear whether each bioactive region is its own isoform or part of a larger mature peptide. To account for this and to possibly discover novel but perhaps not naturally occurring putative AMPs, cleaved peptides were also recombined in a manner similar to alternative splicing (Appendix A). In this procedure, the order and orientation of the cleaved peptides were maintained, and cleaved peptides that originally share cleavage sites were not recombined, with a maximum of three cleaved peptides within recombination. This recombination feature can be turned off in rAMPage’s options.

The collected candidate peptide sequences were classified with AMPlify v1.0.3 [27] as AMP or non-AMP sequences. When given a sequence, AMPlify calculates a score between 0 to 80, with the score ≈ 3.0103 corresponding to the classification probability cutoff of 50% through Equation (1).

To facilitate AMP synthesis for the validation experiments, we filtered the putative AMPs by length and charge, in addition to the AMPlify score. A maximum length of 30 AA was imposed to control the cost of peptide synthesis and to reduce the number of spurious hits from recombined sequences. A minimum charge of +2 was imposed as a proxy to assess the effectiveness of an AMP, as past evidence indicates that more positively charged AMPs show higher activity, especially when their mechanism of action is membrane disruption [69]. Because AMPlify was trained on mostly amphibian AMPs, different score thresholds were imposed for amphibian (≥10) and insect (≥7) datasets to compensate for the dearth of insect training AMPs.

To annotate the final set of filtered putative AMPs, E_N_TAP v0.10.7, Eukaryotic Non-Model Transcriptome Annotation Pipeline [70], were used, along with UniProtKB (release 2020_06, accessed on 15 December 2020) [71], RefSeq (release 203, accessed on 15 December 2020) [72], and NCBI non-redundant (nr) (v5, accessed on 12 December 2020) [29] protein databases. For AMPs that E_N_TAP failed to annotate, InterProScan 5 v5.30-69.0 [37] was run separately to annotate protein families, functions, and domains. Exonerate v2.4.0 [73] was used to align the filtered putative AMPs against the reference AMPs to assess how many of the labeled AMPs were already known AMPs. Finally, SABLE v4.0 [74] was optionally used to predict secondary structures of the filtered putative AMPs, for visualization.

### 4.4. Selecting Filtered Putative AMPs for Validation

To select peptides to validate from the filtered putative AMPs, we ranked their sequences using AMPlify and chose peptides based upon three selection criteria (Figure 3): “Species Count” (*n* = 7), “Insect Peptide” (*n* = 12), or “AMPlify Score” (*n* = 2), for a final total of 21 AMPs (Appendix A). The sequences were first clustered using CD-HIT [75] v4.8.1 with a sequence similarity cutoff of 100%. We chose the longest sequence for each of these clusters, removing duplicate and subsumed sequences to obtain a non-redundant sequence set.

In the first selection strategy of “Species Count”, sequences that were present in more than two species were chosen. In the “Insect Peptide” strategy, to balance the training bias of AMPlify towards AMPs of amphibian origin, we specifically selected insect-originating sequences using a reduced AMPlify score cutoff of >20. In the “AMPlify Score” strategy, the two highest-scoring peptides (AMPlify score = 80.0, 69.9) with the highest charge (+4) were chosen for validation.

### 4.5. Antimicrobial Susceptibility Testing (AST)

Twenty-one putative AMP sequences identified using the rAMPage pipeline were validated through a minimum of three AST experiments performed independently on separate days. In these tests, the AMP activity was assessed using two metrics: minimum inhibitory concentration and minimum bactericidal concentration (MIC and MBC, respectively). MIC and MBC values were determined using procedures outlined by the Clinical and Laboratory Standards Institute (CLSI), with the recommended adaptations for the testing of cationic AMPs described previously [76]. “Wild-type” strains of *Escherichia coli* (*E. coli* 25922) and *Staphylococcus aureus* (*S. aureus* 29213) were purchased from the American Type Culture Collection (ATCC; Manassas, VA, USA) and were used for screening of antimicrobial activity. Briefly, putative AMPs were synthesized by Genscript (Piscataway, NJ, USA) and received in lyophilized format. These peptides were suspended using ultrapure water (Life Technologies, Grand Island, NY, USA; Invitrogen cat# 10977-015), and an 11 μL two-fold serial dilution of 1280 to 2.5 μg/mL was prepared in duplicate rows in a 96-well microtiter plate, before being combined with 100 μL standardized bacterial inoculum yielding a final duplicate testing range of 128 to 0.25 μg/mL. The bacterial inoculum was prepared using colonies isolated on non-selective agar and combined with Mueller Hinton Broth. This suspension was measured and adjusted to achieve an optical density of 0.08–0.1, equivalent to a 0.5 McFarland standard (1–2 × 10^8^ cfu/mL). The inoculum was then diluted to a target concentration of 5 ± 3 × 10^5^ cfu/mL; total viable counts from the final inoculum were routinely performed to confirm the target bacterial density was achieved. MIC values were reported at the concentrations in which no visible growth was detected following 20–24-h incubation at 37 °C. The MIC and adjacent wells were plated onto non-selective agar; the concentration in which killed 99.9% of the inoculum following additional overnight incubation was determined to be the MBC.

### 4.6. Hemolysis Experiments

The twenty-one putative AMPs were evaluated for toxicity using three independent hemolysis experiments performed on separate days. Whole blood from healthy donor pigs was purchased from Lampire Biological Laboratories (Pipersville, PA, USA). Red blood cells (RBCs) were washed and isolated by centrifugation, using Roswell Park Memorial Institute medium (RPMI) (Life Technologies, Grand Island, NY, USA; Gibco cat# 11835-030). Lyophilized AMPs were suspended and serially diluted from 128–1 μg/mL using RPMI in a 96-well plate, before being combined with 100 μL of a 1% RBC solution. Following a minimum 30 min incubation at 37 °C, plates were centrifuged and ½ volume from each supernatant was transferred to a new 96-well plate. The absorbance of these wells was measured at 415 nm. To quantify hemolytic activity and determine the AMP concentration that kills 50% of the RBCs (HC_50_), absorbance readings from wells containing RBCs treated with 11 μL of a 2% Triton-X100 solution or RPMI (AMP solvent-only) were used to define 100% and 0% hemolysis, respectively. All centrifugation steps were performed at 500× *g* for five minutes in an Allegra-6R centrifuge (Beckman Coulter, CA, USA).

## 5. Conclusions

rAMPage is a bioinformatics pipeline for high throughput identification of putative AMPs in RNA-seq datasets. It fills a current void in the AMP discovery process, bridging the gap between in silico and in vitro methods. The pipeline has the potential to accelerate the discovery of novel antibiotics, with the possibility to enrich existing AMP sequence repositories. The easy-to-run pipeline design with various checkpoints and the low computational resources required to run rAMPage increase its accessibility to users. By executing rAMPage on publicly available amphibian and insect transcriptome sequencing data, we have identified over 1000 putative AMPs. Of those, we performed functional tests on twenty-one putative AMPs and demonstrated that seven have moderate to high activity against *E. coli* ATCC 25922 and/or *S. aureus* ATCC 29213. As the number of tested peptides increases, the wet lab validation results can feed back into rAMPage by augmenting the reference AMP datasets, helping refine the underlying homology and machine learning approaches. We expect rAMPage to have broad utility in the discovery of novel antimicrobials from a wide variety of transcriptome sequencing datasets.

## 6. Patents

Patent applications pending on the reported novel peptides.

## Figures and Tables

**Figure 1 antibiotics-11-00952-f001:**
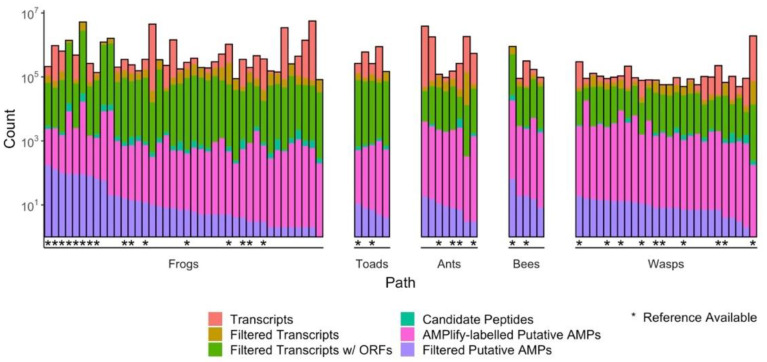
Statistics and attrition as the sequencing data are processed by the rAMPage AMP discovery pipeline. rAMPage processes RNA-seq datasets from raw reads to transcripts to putative AMPs. In this case, a putative AMP is defined as a sequence with an AMPlify score ≥10 for amphibians or ≥ 7 for insects, a length ≤ 30 AA, and a charge ≥ 2. Datasets with a reference transcriptome used during assembly are indicated with an asterisk. The total number of putative AMPs (*n* = 1478, including 341 duplicates) are shown in purple, discovered from a total of ~53 million assembled transcripts.

**Figure 2 antibiotics-11-00952-f002:**
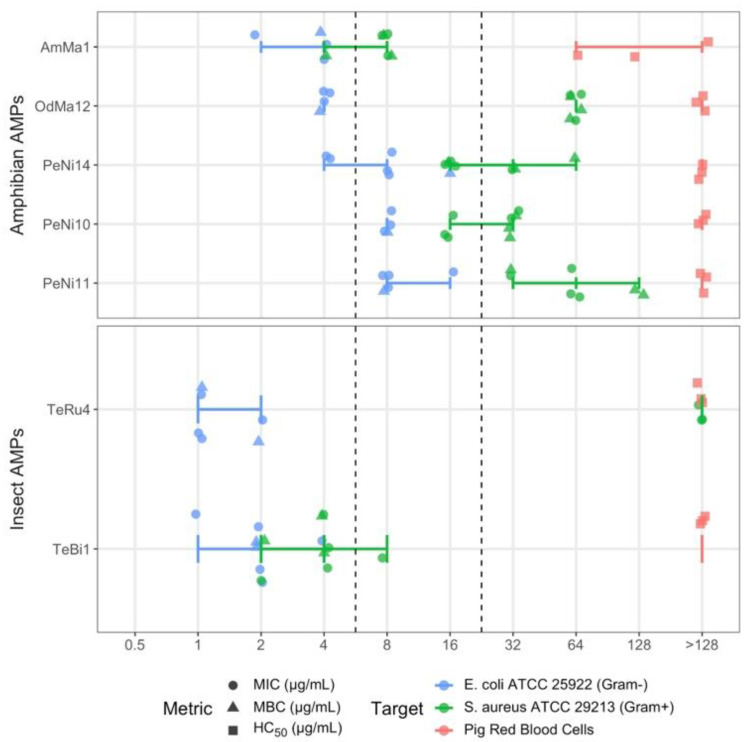
Antimicrobial susceptibility and hemolysis test results of seven moderately and highly active putative AMPs. AMPs were tested for their bioactivity against *E. coli* and *S. aureus* to determine minimum inhibitory and bactericidal concentrations (MIC and MBC, respectively). AMPs were also tested for their hemolytic activity using pig red blood cells to determine hemolytic concentration (HC_50_) values. Moderate activity (MIC and MBC in the range of 8–16 μg/mL) and high activity (≤4 μg/mL) thresholds indicated by the dashed lines. AMPs are ordered by increasing MIC values against *E. coli* ATCC 25922.

**Figure 3 antibiotics-11-00952-f003:**
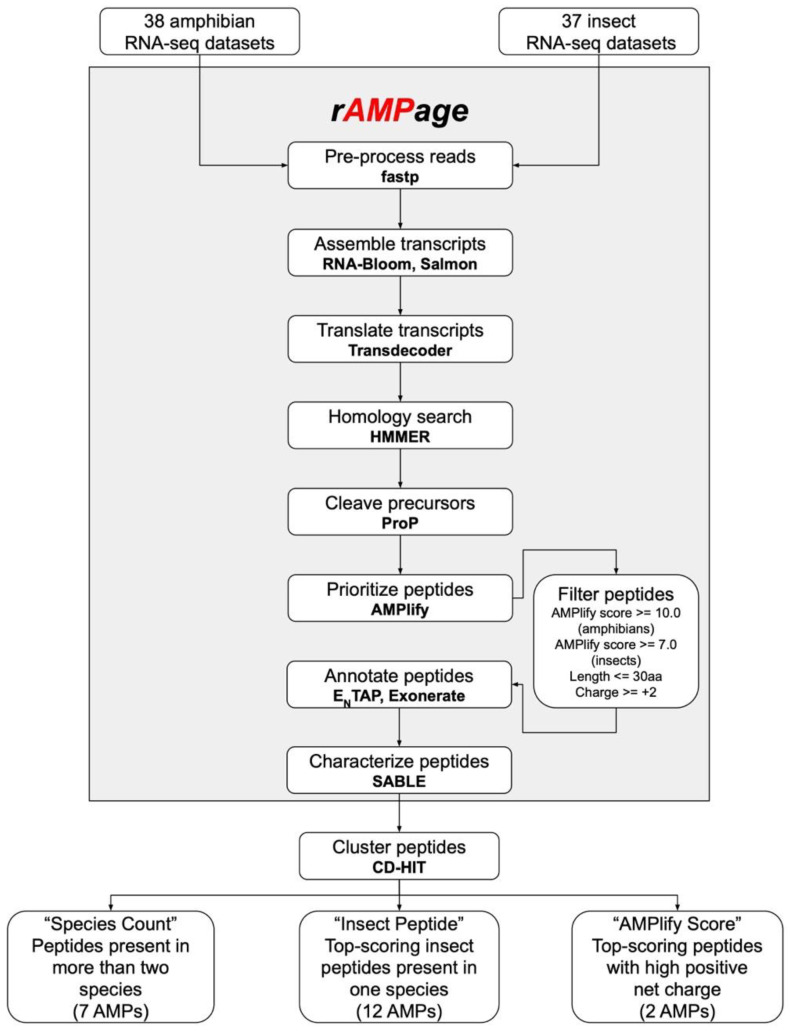
rAMPage workflow. The rAMPage pipeline and downstream selection of putative AMPs for validation.

**Table 1 antibiotics-11-00952-t001:** Characteristics of putative AMP sequences with moderate to high in vitro bioactivity against *E. coli* or *S. aureus*. Each sequence is separated into the prepro sequence and the predicted mature peptide sequence. Conserved proteolytic cleavage sites are underlined in the prepro sequences.

Prepro-Sequence	Putative Mature Peptide						
	Sequence	Length	Charge	AMPlify Score	MIC (μg/mL) *		Peptide ID
					E. coli ^†^	*S. aureus* ^†^	
MFTMKKSLLVLFFLGIVSLSLCEEERNADEDDGEMTEEVKR	GILDTLKQLGKAAVQGLLSKAACKLAKTC	29	4	80.0	2–4	4–8	AmMa1
LGIVSLSLCQEERSADDEEGEVIEEEVKR	GFMDTAKNVAKNVAVTLLYNLKCKITKAC	29	4	69.2	4	64	OdMa12
MFTMKKSLLFFFLGTIALSLCEEERGADEEENGGEITDEEVKR	GLLLDTVKGAAKNVAGILLNKLKCKVTGDC	30	3	61.8	8	16–32	PeNi10
MFTMKKSLLLVFFLGTIALSLCEEERGADDDNGGEITDEEIKR	GILTDTLKGAAKNVAGVLLDKLKCKITGGC	30	3	61.8	8–16	32–128	PeNi11
MFTLRKSLLLLFFLGMVSLSLCEQERDADEDEGEVTEEVKR	GLWTTIKEGVKNFSVGVLDKIRCKITGGC	29	3	67.5	4–8	16–64	PeNi14
MKLLALVLVLSCVVAYTTARKRGQYWPTNTKIFTTPYRFRREADQGSIVANLKNTPQLPFDDNENLRLVLFDNDPTVDLGEDDKEIPGPQSQPNALSNNLHLIDENDYFSSYTSQPGTYRSFPRNFGTSGRYRWRREAGGHVEPRLRFDAETQRGNSFFTDFADLQRRANGRGIEPTVSATAGIRFRQEADQINPLAVRRERR	SWLSKSVKKLVNKKNYTRLEKLAKKKLFNE	30	8	25.5	1–2	>128	TeRu4
IFLVGCKLFGNFILQRMQLLLALADAVA	KIKIPWGKVKDFLVGGMKAVGKK	23	6	45.0	1–4	2–8	TeBi1

* MIC: Minimum inhibitory concentration. ^†^
*Escherichia coli* ATCC 25922; *Staphylococcus aureus* ATCC 29213.

**Table 2 antibiotics-11-00952-t002:** Comparison of sequence identities (%) of the discovered AMPs with their best-known AMP blastp matches to the NCBI non-redundant (nr) protein database over the entire sequence (precursor), prepro or mature sequences.

Peptide ID	Source Organism	Highest Scoring Blastp Match	Sequence Identity (%)		
			Precursor	Prepro	Mature
AmMa1	*Amolops mantzorum*	Palustrin-2GN3 (ADM34231.1) [*Amolops granulosus*]	97	100	93
OdMa12	*Odorrana margaretae*	Odorranain-F2 (ABG76517.1) [*Odorrana grahami*]	98	100	97
PeNi10	*Leptobrachium boringii* *Polypedates megacephalus* *Pelophylax nigromaculatus* *Rhacophorus dennysi* *Rhacophorus omeimontis*	Pelophylaxin-1 (Q2WCN8.1) [*Pelophylax fukienensis*] Ranatuerin-2N (AEM68233.1) * [*Pelophylax nigromaculatus*]	82 98	86 97	77 100
PeNi11	*Leptobrachium boringii* *Polypedates megacephalus* *Pelophylax nigromaculatus* *Rhacophorus dennysi* *Rhacophorus omeimontis*	Pelophylaxin-1 (Q2WCN8.1) [*Pelophylax fukienensis*]	100	100	100
PeNi14	*Bufo gargarizans* *Polypedates megacephalus* *Pelophylax nigromaculatus* *Rhacophorus omeimontis*	Palustrin-2HB1 (AIU998997.1) [*Pelophylax hubeiensis*]	90	93	86
TeRu4	*Temnothorax rugatulus*	Uncharacterized protein (XP_024884948.1) [*Temnothorax curvispinosus*] Uncharacterized protein (TGZ47385.1) * [*Temnothorax longispinosus*]	94 91	93 90	97 97
TeBi1	*Tetramorium bicarinatum*	M-myrmicitoxin(01)-Tb1a (W8GNV3.1) [*Tetramorium bicarinatum*]	100	-	100

* Highest scoring blastp match when query sequence consists of only the mature sequence instead of the whole precursor. -: no significant alignment.

## Data Availability

Accessions for input RNA-seq datasets can be found in Appendix A. rAMPage code is publicly available at https://github.com/bcgsc/rAMPage (v1.0, accessed on 14 February 2021).

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
