# Peer review of "Mining Amphibian and Insect Transcriptomes for Antimicrobial Peptide Sequences with rAMPage"

_antibiotics, 2022, doi:10.3390/antibiotics11070952_

Round 1

Reviewer 1 Report

In this manuscript, authors have developed a scalable bioinformatics discovery platform namely, rAMPage, for the discovery of AMP sequences in RNA-Seq datasets available in public domain. I hope this platform may be useful in future to discover novel antimicrobial peptides. 

It is a well-written manuscript.
Introduction has been written very well. Methods are well explained
Results and figures are well explained.
Discussion and conclusion explain very well the outcomes of this manuscript.

Author Response

We thank for our reviewer's support.

Reviewer 2 Report

For many years AMPs have been suggested as alternatives to antibiotics in order to counteract resistance. While many AMPs are being identified routinely, there exist many roadblocks exist to their clinical use due to poor pharmacokinetic properties and side effects. So additional methods to identify newer AMPs from existing databases is a very worthy investigation. The relatively low computation requirement of the rAMPage algorithm is commendable, making it a very accessible technique for mining various existing databases. The scientific merit of the work in this paper definitely warrants publication. 

This being declared, the paper requires extensive proofreading. There are many typographical errors (e.g Line 147  Lower MIC and MBC values are desirable as they indicate that lover lower AMP) and there are two figures in the manuscript labeled as Figure 2. Elementary proofreading errors like this make the manuscript appear to be sloppy.

One additional issue is that the Discussion section is written in a very introspective manner. The authors have to also consider other existing approaches in the field. There are many existing computational methods to identify AMPs. The authors have to discuss what distinguishes them from existing algorithms and discuss pros and cons objectively. Having multiple approaches to address a given problem is an advantage in itself and the low computational requirement is definitely a positive. 

The manuscript is recommended for major revisions due to the extensive rewriting that is required to make it publishable.

Author Response

This being declared, the paper requires extensive proofreading. There are many typographical errors (e.g Line 147  Lower MIC and MBC values are desirable as they indicate that lover lower AMP) and there are two figures in the manuscript labeled as Figure 2. Elementary proofreading errors like this make the manuscript appear to be sloppy.

[Response] The mentioned typo on former Line 147 has been corrected. We have also proofread the manuscript further and made other minor corrections to enhance clarity of the manuscript. We will pay closer attention during submission to make sure that the production file is rendered correctly for figure numbers.

One additional issue is that the Discussion section is written in a very introspective manner. The authors have to also consider other existing approaches in the field. There are many existing computational methods to identify AMPs. The authors have to discuss what distinguishes them from existing algorithms and discuss pros and cons objectively. Having multiple approaches to address a given problem is an advantage in itself and the low computational requirement is definitely a positive.

[Response] We have added a paragraph discussing other computational methods in the field, and discussed our choice of employing AMPlify in the rAMPage pipeline. Additionally, a brief review of the existing methods in the field was written in the Introduction.

Reviewer 3 Report

This manuscript describes the mining amphibian and insect transcriptomes for antimicrobial peptide sequences with in silico rAMPage methodology, with that, able to identify novel putative Antimicrobial peptides.  As the authors highlighted, the anti-microbial resistance of microorganisms is becoming an increasingly serious concern in medicine, agriculture, and animal husbandry, especially in developing countries.  I believe this type of study is important to combat the antimicrobial therapeutics area.  The manuscript is well organized.

Lines 132 to 134: I do not know what is relevant with the timeline?

Introduction – Overall written very well.

Result – Most of the earlier study library was prepared using poly using poly(A) enrichment of the mRNA (mRNA-Seq).  Since bacteria don’t have poly(A), technically, there is no bacterial mRNA (mRNA-Seq) in the transcriptomes.  But many studies have shown that bacteria can also produce antimicrobial peptides.  Any though?

Lines 145  to 147:  “In these antimicrobial ……….. (MIC and MBC, respectively)” this should be a move to method sections.

Line 176: “ (sequence identity = 100%)” what was the query coverage used in to determine?  Top hit with identity = 100%, but the query has 5 % or 10%, does not mean anything.  Clarifying how this was carried out.

Discussion – Some earlier studies showed that Escherichia coli primed by sub-lethal doses of AMPs develop tolerance and increase persistence by creating curli or colanic acid and biofilm formation (see https://doi.org/10.1371/journal.ppat.1009443).  I suggest this concern also needs to be included in the discussion.  

Line 407 to 428:  How many times was this replicated?  All the tests were done on was day or at different times?

Line 429 to 440: This experiment also lacks the replicates details. 

Author Response

Lines 132 to 134: I do not know what is relevant with the timeline?

[Response] This line is to exhibit the low computational and time resources required to run the rAMPage pipeline, making it accessible to research labs and users without large computing resources and clusters. To make this more relevant with the flow of our narrative, we moved the statement to the end of the first paragraph of the same section, and renumbered our Supplementary Figures 1 and 2 accordingly. This point is also reiterated in the Conclusion.

Result – Most of the earlier study library was prepared using poly using poly(A) enrichment of the mRNA (mRNA-Seq).  Since bacteria don’t have poly(A), technically, there is no bacterial mRNA (mRNA-Seq) in the transcriptomes.  But many studies have shown that bacteria can also produce antimicrobial peptides.  Any though?

[Response] We thank our reviewer for pointing out this interesting use case. While the reported work is focused on applying rAMPage to mining amphibian and insect transcriptomes (as our title suggests), it is also possible to explore bacterial transcriptomes using a similar strategy. Although we have not explicitly tested this particular use case, we have added some text in our Discussion section how this can be done.

Lines 145  to 147:  “In these antimicrobial ……….. (MIC and MBC, respectively)” this should be a move to method sections.

[Response] Because MIC and MBC need to be defined while presenting our results, this line has been kept where it is. However, the terms are also defined now in the Methods section, as suggested.

Line 176: “ (sequence identity = 100%)” what was the query coverage used in to determine?  Top hit with identity = 100%, but the query has 5 % or 10%, does not mean anything.  Clarifying how this was carried out.

[Response] We thank our reviewer for asking for clarification on this. Here, we are checking to see if any of our discovered sequences exist in the NCBI nr database in its current form (including its current length). In other words, peptides are considered novel if they do not have a corresponding entry in the database with a sequence identity = 100% and query coverage = 100%. This is because even small mutations or truncations in peptide sequences may impact their antimicrobial activity. To clarify, we have added the required query coverage percentage on this line.

Discussion – Some earlier studies showed that Escherichia coli primed by sub-lethal doses of AMPs develop tolerance and increase persistence by creating curli or colanic acid and biofilm formation (see https://doi.org/10.1371/journal.ppat.1009443). I suggest this concern also needs to be included in the discussion. 

[Response] We have added text regarding resistance to AMPs in bacteria in the Discussion section, as suggested.

Line 407 to 428:  How many times was this replicated?  All the tests were done on was day or at different times?

[Response] All experiments were replicated three times on different days. We have added this information in the text.

Line 429 to 440: This experiment also lacks the replicates details.

[Response] This experiment was also done in three replicates. We have added this information in the text.

Round 2

Reviewer 2 Report

The revised version has adequately addressed comments in the previous round of review and hence is recommended for publication.